



# The Prevalence of Precipitation from Polar Supercooled Clouds

Israel Silber[1], Ann M. Fridlind[2], Johannes Verlinde[1], Andrew S. Ackerman[2], Grégory V. Cesana[2,3,] Daniel A. Knopf[4]

[1] Department of Meteorology and Atmospheric Science, Pennsylvania State University, University Park, PA, USA
[2] NASA Goddard Institute for Space Studies, New York, NY, USA
[3] Center for Climate Systems Research, Earth Institute, Columbia University, New York, NY, USA
[4] School of Marine and Atmospheric Sciences, Stony Brook University, Stony Brook, NY, USA

*Correspondence to*: Israel Silber (ixs34@psu.edu)

**Abstract.** Supercooled clouds substantially impact polar surface energy budgets but large-scale models often underestimate
their occurrence, which motivates accurately establishing metrics of basic processes. An analysis of long-term measurements
at Utqiaġvik, Alaska, and McMurdo Station, Antarctica, combines lidar-validated use of soundings to identify supercooled
cloud layers and colocated ground-based profiling radar measurements to quantify cloud base precipitation. We find that more
than 85% (75%) of sampled supercooled layers are precipitating over the Arctic (Antarctic) site, with more than 75% (50%)
precipitating continuously to the surface. Such high frequencies can be reconciled with substantially lesser spaceborne
estimates by considering differences in radar hydrometeor detection sensitivity. While ice precipitation into supercooled clouds
from aloft is common, we also find that the great majority of supercooled cloud layers without ice falling into them are
themselves continuously generating precipitation. Such sustained primary ice formation is consistent with continuous
activation of immersion-mode ice nucleating particles (INPs), suggesting that supercooled cloud formation is a principal
gateway to ice formation at temperatures greater than ~–38 °C over polar regions. The prevalence of weak precipitation fluxes
is also consistent with supercooled cloud longevity, and with well-observed and widely simulated case studies. An analysis of
colocated microwave radiometer retrievals suggests that weak precipitation fluxes can be nonetheless consequential to
moisture budgets for supercooled clouds owing to small liquid water paths. Finally, we suggest that these ground-based
precipitation rate statistics offer valuable guidance for improving the representation of polar cloud processes in large-scale
models.

## 1 Introduction

Supercooled clouds exert substantial radiative impacts on the surface energy budget over polar regions (e.g., Dong et al.,
2010; Miller et al., 2015; Shupe & Intrieri, 2004; Silber et al., 2019a), and play an important role in Arctic amplification and
solar absorption over the Southern Ocean (e.g., Cronin & Tziperman, 2015; McCoy et al., 2014, 2015; Pithan et al., 2018; Tan
& Storelvmo, 2019). However, major uncertainties in their representation in climate models (e.g., Cesana et al., 2015;



Tsushima et al., 2006) ensue from knowledge gaps concerning the active processes affecting cloud lifecycles (e.g., Tan et al., 2016; Kay et al., 2018).

Both ice and liquid precipitation can form in supercooled clouds. At temperatures warmer than the homogeneous freezing of water (~–38°C), ice initiation typically requires ice-nucleating particles (INP; e.g., Hegg & Baker, 2009; Vali et al., 2015),
and it may be enhanced by secondary processes such as rime-splintering or droplet shattering (e.g., Hallett & Mossop, 1974; Korolev et al., 2020). Once formed, ice hydrometeors grow rapidly by depositional growth both within supercooled cloud and in underlying ice-supersaturated layers (e.g., Pinsky et al., 2015), and by collisions with droplets (riming) and other ice hydrometeors (aggregation) (e.g., Fridlind & Ackerman, 2018). Less commonly formed supercooled drizzle drops grow only within supercooled cloud primarily by accretion of cloud droplets (e.g., Rangno & Hobbs, 2001).

Precipitation impacts the lifecycle of supercooled clouds even if the cloud base flux is weak, or if it evaporates or sublimates before reaching the surface (e.g., Silber et al., 2019d; Solomon et al., 2011). However, few studies have quantified precipitation occurrence from polar supercooled clouds. McIlhattan et al. (2017) reported that ~10% of shallow supercooled clouds are precipitating to the surface, based on Arctic-wide retrievals from the Cloud Profiling Radar (CPR; Tanelli et al., 2008) onboard CloudSat (Stephens et al., 2002). Edel et al. (2020) reported a greater value (~20%) by including a lower likelihood of CPR
surface precipitation (see Wood, 2011). Also based on CPR measurements, Zhang et al. (2010) found that ~60% of polar mid-level supercooled clouds are precipitating at or below cloud base. By contrast, ground-based remote-sensing measurements over the Arctic commonly show essentially continuous precipitation from supercooled cloud decks (e.g., Fridlind & Ackerman, 2018, Fig. 3; Shupe, 2011; Shupe et al., 2006), generally commencing shortly after supercooled cloud formation (e.g., de Boer et al. 2011).

Here we attempt to reconcile a general impression from ground-based measurements that polar supercooled clouds are nearly universally precipitating with a quite wide range of satellite-based estimates. Using multiple years of sounding and closely colocated ground-based radar observations from Arctic and Antarctic sites, we first evaluate the occurrence of cloud base and surface precipitation from all detected supercooled cloud layers. We then examine the impact of radar sensitivity on precipitation detectability, and derive estimates of cloud base precipitation flux. Finally, for single-layer clouds, we provide a
rough measure of precipitation impact on cloud lifetime by comparing the estimated cloud base precipitation rates with simultaneous microwave radiometer retrievals of liquid water path.

## 2 Methodology

To detect supercooled cloud layers, we use 6 or 12 hourly soundings acquired at Utqiaġvik (formerly Barrow), North Slope of Alaska (NSA; Verlinde et al., 2016) from November 2011 to April 2019 and 12 hourly soundings acquired at McMurdo
Station, Antarctica between December 2015 and January 2017 (Lubin et al., 2020). After linearly interpolating onto a 15 m



grid, supercooled layers are identified where atmospheric temperature is between 0 and -40 °C and relative humidity (RH) exceeds 95% over at least two adjacent grid cells, consistent with an RH uncertainty of 5% (Holdridge et al., 2011). This method shows good agreement (in more than 90% of cases) with independent retrievals from lidar measurements (see Silber et al., 2020b, Fig. S1) and permits layer identification over the full column, which is not limited by lidar extinction.

To detect precipitation, we use colocated Ka-band zenith-pointing radar (KAZR; Widener et al., 2012) measurements acquired above 300 m ($h_{min}$). After interpolating onto the same 15 m grid, hydrometeor-containing range gates are taken as those where radar echoes exceed the noise floor (see Silber et al., 2018a) in at least 50% of profiles within 15 min after the radiosonde release. We estimate likely biases resulting from the binary averaging (e.g., Smalley et al., 2014) on this occurrence percentage threshold to be below 10% (not shown). In using a 15-min window we account for sheared fall streak structures

while the short duration mitigates biases during rapid changes in liquid layer height (not shown; e.g., Verlinde et al., 2013, Fig. 3). We omit supercooled layers below $h_{min}$ and above 4.3 km, which is the average altitude reached within 15 min of radiosonde release.

KAZR equivalent reflectivity factor ($Z_e$) and mean Doppler velocity ($V_D$) profiles are arithmetically averaged over hydrometeor-containing volumes within each 15-min window. While the sounding-based supercooled cloud detection method

is powerful for establishing cloud boundaries, it unfortunately does not allow us to establish the cloud occurrence fraction profile over the 15-min window. The averaging period selected for $Z_e$ and $V_D$ could result in a potential bias of $V_D$ and either bias or overestimation of $Z_e$, depending on whether the full window or the hydrometeor-containing range gate occurrence fraction is used, respectively. Here we omit hydrometeor-free samples from these calculations, resulting in a potential overestimation that is less than 0.5 (0.6) dBZ in more than 90% of the cases over the NSA (McMurdo), based on hydrometeor-

containing range gate occurrence fraction statistics (not shown). Our statistics and conclusions are therefore not sensitive to this choice.

To define precipitation occurrence we require that mean $Z_e$ over a fixed depth below liquid cloud base ($d_{min}$) exceed a fixed threshold ($Z_{e_{min}}$). In support of that, positive $V_D$ values (towards the surface) just below cloud base in more than 95% of the hydrometeor containing range gates in both datasets indicate the presence of precipitating hydrometeors. By varying $d_{min}$ and

$Z_{e_{min}}$ we emulate different range resolutions and instrument or algorithm sensitivity thresholds. The smallest $Z_{e_{min}}$ (–50 dBZ) corresponds to the KAZR sensitivity at ~1 km (see Appendix A), with an effective uncertainty of several dBZ (e.g., Kollias et al., 2019). The smallest $d_{min}$ (60 m) corresponds to two KAZR range gates. Supercooled cloud layers extending below $h_{min}$ + $d_{min}$ are omitted because the cloud base is not discernable by KAZR (below $h_{min}$) or an emulation with coarsened vertical resolution (below $h_{min}$ + $d_{min}$). The resulting McMurdo dataset constitutes 236 ($d_{min}$ = 600 m) to 262 ($d_{min}$ = 60 m) profiles with

at least one supercooled layer whereas the NSA dataset constitutes 3,139 to 4,544 profiles, reflecting a higher occurrence of low-level supercooled clouds over the NSA (e.g., Lubin et al., 2020, Fig. 7).





To evaluate surface precipitation occurrence, we compare linearly averaged $Z_e$ at $h_{min}$ to $Z_{e_{min}}$ in profiles with at least one supercooled cloud layer. The impact of ground-based versus space-borne effective $h_{min}$ (typically ~0.3 and ~1.2 km, respectively) on surface precipitation occurrence was estimated at both sites to be roughly ±10 percentage points, suggesting

hydrometeor nucleation, growth, evaporation, or sublimation within this "blind zone" (not shown; cf. Bennartz et al., 2019; Castellani et al., 2015; Maahn et al., 2014). We note that the impact of the blind zone between the surface and ground-based $h_{min}$ on precipitation occurrence using high radar sensitivities, similar to those used here, is absent from the literature to our knowledge and merits a dedicated study.

To estimate precipitation rate ($R$) immediately below cloud base ($R_{CB}$), following Bühl et al. (2016) and because a reliable

retrieval of mass-weighted fall speed is not available, we simply multiply $V_D$ by retrieved ice water content (IWC) following Hogan et al. (2006). Using this method, we do not apply any ice habit property constraints on the observations, which span the full heterogeneous freezing temperature range, but do find some overlap with various $Z_e$-$R$ relationship parametrizations (see Appendix B). We roughly estimate the IWC uncertainty as -90% to +300%, based on the range of retrieval errors deduced by Heymsfield et al. (2008), by which we implicitly consider potentially different prevailing ice properties from the measurements

used in their study. We neglect $V_D$ uncertainty (~0.1 m/s; Widener et al., 2012) since it is comparatively negligible, e.g., ±10% when considering the distribution of $V_D$ values over the NSA (see Silber et al., 2020a). Similarly, we neglect the impact of short-term air-motion variability over $V_D$ because it is largely canceled by the $V_D$ averaging over the 15-min window, based on a comparison with averaging calculations using 1-h windows. This comparison resulted in similar 15-min and 1-h averaged $V_D$ distributions with a mean difference of 1.3 and 4.5% at the NSA and McMurdo, respectively (not shown; see Sedlar &

Shupe, 2014; Shupe et al., 2008b, 2008a).

We use microwave radiometer (MWR: Morris, 2006) retrievals of liquid water path (LWP; Cadeddu et al., 2007; Turner et al., 2007) for single-layer profiles (53 and 60% of all cloud-containing profiles over NSA and McMurdo, respectively). We do not limit LWP to values greater than the widely-used uncertainty of 25 g/m² (e.g., Turner et al., 2007; Westwater et al., 2001) because doing so would exclude frequently occurring tenuous supercooled clouds (e.g., Sedlar, 2014; Silber et al.,

2020b) which account for 32% (73%) of the NSA (McMurdo) single-layer profiles (shown below). We note that the MWR LWP retrievals predominantly exhibit significantly smaller errors, averaging around 0 g/m² in bulk statistics (no retrieval bias; see Cadeddu et al., 2009, 2013).

## 3 Precipitation Statistics

Based on KAZR measurement capabilities (minimum cloud depth and $Z_e$ thresholds of 60 m and -50 dBZ, respectively),

we find that more than 85% (75%) of all sampled supercooled cloud layers over the NSA (McMurdo) precipitate from liquid cloud base (Figure 1a,b). Here each supercooled layer is counted separately over columns that contain both single- and multi-



layer cases. We interpret these percentages as lower limits on precipitation occurrence. In some cases, KAZR sensitivity appears to be a limiting factor or precipitating ice does not grow large enough to be detected by the KAZR immediately below cloud base (see Appendix C). Such cases may explain the distribution of non-precipitating cloud top temperatures over the

NSA reaching a minimum, corresponding to a maximum likelihood of precipitation detection, at -15 °C and (to a lesser extent) -5 °C (Figure 2a), where vapor growth rate peaks (e.g., Fukuta & Takahashi, 1999). Similarly, shallow clouds (Figure 2c) or clouds at the low end of LWP (Figure 2d), the cases of which frequently overlap (not shown), may hamper ice growth to detectable sizes by limiting the time ice particles can grow via vapor deposition or riming during sedimentation from the (coldest) cloud top regions where INP activation is expected to be strongest.

Over the NSA, where statistics are most robust, cloud base precipitation fraction remains 0.8 or higher throughout the heterogeneous freezing regime (Figure 2a). Such high fractions are in part influenced by the commonality of seeding from overlying precipitation falling into supercooled cloud tops (primarily ice-phase precipitation, as discussed below), which occurs in 47% (45%) of sampled supercooled layers over the NSA (McMurdo) (not shown; cf. Vassel et al., 2019). However, when only the topmost supercooled layers with no overlying precipitation are considered, the percentages are reduced by only

roughly 10% (Figure 1c,d), suggesting that supercooled clouds are usually generating precipitation.

Surface precipitation occurrence in supercooled cloud-containing profiles is greater than 75% (50%) over the NSA (McMurdo) (Figure 1e,f). Here each column is counted as a single case, whether it contains one or more supercooled layers, in order to remain comparable with spaceborne statistics that use lidar measurements to detect at least one supercooled layer in a column and radar measurements to detect underlying surface precipitation. The lower percentage over McMurdo may be

influenced by intense near-surface sublimation augmented by katabatic winds (e.g., Grazioli et al., 2017). The precipitation detected with KAZR may be liquid or ice phase. However, since in these datasets $Z_e$ usually increases from cloud base to some distance below (not shown), indicating continued ice growth during sedimentation rather than drizzle or rain evaporation, we infer that ice is the dominant precipitation form (e.g., Edel et al., 2020; Rangno & Hobbs, 2001; Shupe, 2011).

Continuous precipitation of ice from non-seeded supercooled cloud layers suggests continuous in-cloud activation of INP

(e.g., de Boer et al., 2011; Westbrook & Illingworth, 2013). Because INP availability generally increases exponentially with decreasing temperature (e.g., DeMott et al., 2010), we posit that longwave radiative cooling is the primary driver of continuous activation of INP initially present in a cloud layer. We note that over such high-latitude regions the cloud top longwave radiative cooling typically remains significantly greater than shortwave radiative heating throughout sunlit periods (e.g., Turner et al., 2018). In the roughly three-quarters of cases where cloud layers are turbulent (Silber et al., 2020b), additional INP may

be continuously entrained at cloud-top (e.g., Fridlind et al., 2012) and potentially at cloud base via deepening of a decoupled layer (e.g., Avramov et al., 2011). In non-turbulent layers, progressive saturation of an increasing cloud depth (e.g., Silber et al., 2020b) could also effectively increase the in-cloud INP pool. The overall differences in detected Arctic versus Antarctic



precipitation frequency (Figure 1) are likely influenced by geographical INP variability associated with both long-range transport and local source regions (e.g., Vergara-Temprado et al., 2018).

Profiles of INP or aerosol properties are unfortunately not retrievable from the available McMurdo and NSA measurements, but we can establish the degree to which non-precipitating cases may exhibit conditions that would likely be associated with a scarcity of activatable INP relative to all clouds for the 7-year NSA dataset. For instance, non-precipitating clouds are more common at temperatures closer to 0 °C, where activation of INP is known to be extremely scarce (bars in Figure 2a; see Rangno & Hobbs, 2001; see also Figures S3 and S5). Non-precipitating clouds also occur mostly during summer (Figure 2b), when

INP and aerosol particle concentrations are lowest (e.g., Fountain & Ohtake, 1985; Quinn et al., 2002, 2009). Third, non-precipitating clouds tend to be thinner and lower in LWP (Figure 2c,d), consistent with slower entrainment. Reduced INP entrainment is also suggested by a statistically significant higher occurrence of non-turbulent clouds being non-precipitating (36%) relative to the full dataset (27%; not shown). Finally, radiative cooling and entrainment of INP may also be suppressed by radiative shielding, consistent with 17% (43%) greater non-precipitating cloud occurrence when adjoining layers are

vertically separated by less than 500 m (100 m) (not shown; cf. Sedlar & Shupe, 2014; see also Figure C3). We found indications of similar non-precipitating case characteristics over McMurdo, but the smaller dataset inhibited a statistically robust analysis.

**4 Reconciling Apparent Precipitation Occurrence**

The detectability of precipitating hydrometeors is a function of the radar characteristics such as operating wavelength,

receiver sensitivity, and pulse width, as well as the spatial characteristics of the precipitation (e.g., Lamer et al., 2019). Here we examine the impact of radar range resolution and $Z_{e_{min}}$ on the reported precipitation percentage by varying these thresholds to emulate these parameters in other radar systems. Results indicate that reducing the radar range resolution can counterintuitively increase the precipitating cloud percentage owing to the higher probability of detecting larger hydrometeors in a larger volume, but higher $Z_{e_{min}}$ can more significantly reduce the cloud and surface precipitation percentages (Figure 1).

For example, emulation of the highest-sensitivity CloudSat CPR $Z_{e_{min}}$ corresponds to surface precipitation percentages that are lower than KAZR by 5-10 percentage points (Figure 1e,f), in agreement with Zhang et al. (2010, their Figs. 6 and 7), who used temperature-dependent $Z_e$ thresholds. Emulation of the $Z_{e_{min}}$ corresponding to the "precipitation possible" flag of the CloudSat 2C-PRECIP-COLUMN (2C-PC; Haynes et al., 2009) and 2C-SNOW-PROFILE (2C-SP; Wood, 2011, ch. 7; Wood et al., 2014) precipitation detection algorithms yields surface and cloud base precipitation occurrences lower than KAZR by

more than 30 (25) points over the NSA (McMurdo) (Figure 1a,b,e,f). Both precipitation occurrences are lower by 15-20 points more when the $Z_{e_{min}}$ corresponding to 2C-PC "solid precipitation certain" flag (-5 dBZ) is emulated (Figure 1a,b,e,f). These





satellite measurement and retrieval sensitivity limitations are accentuated when the Ka-band precipitation radar (KaPR) onboard the Global Precipitation Measurement (GPM) satellite (Hou et al., 2013) sensitivity is adopted for precipitation detection (Figure 1; estimated detection of 1 in 10 precipitation events; cf. Skofronick-Jackson et al., 2019). This result is

consistent with known limitations of the KaPR capability to detect light precipitation (e.g., Hamada and Takayabu, 2016). Aside, we note that the GPM inclination angle of 65° excludes most polar regions, including NSA and McMurdo, but is high enough to observe some relevant high-latitude regions such as the Southern Ocean.

When applying the 2C-PC $Z_{e_{\min}}$ for "certain" and "possible" precipitation (accounting for radar sensitivity and range resolution), our NSA surface precipitation occurrences are still 5–10 points greater than the higher range of central-Arctic

values (~20-40%) estimated by McIlhattan et al. (2017) and Edel et al. (2020), respectively. These remaining differences are likely attributable to $h_{\min}$ differences (see Section 2) and the spatial distribution of Arctic precipitation (relatively higher over the NSA; cf. Edel et al., 2020, Fig. 3; McIlhattan et al., 2017, Fig. 7). Altogether, this radar sensitivity analysis can generally reconcile expected high precipitation occurrence from ground-based measurements with variously lower values derived from satellite data. By emulating the $Z_{e_{\min}}$ and vertical resolution of the future EarthCARE mission's CPR (see Illingworth et al.,

2015; Kollias et al., 2014), we find that it may detect precipitation percentages similar to those of KAZR (Figure 1), thereby better establishing polar precipitation processes.

Finally, we find that the cloud base precipitation occurrence, which is most relevant to cloud lifetimes but currently more challenging to well establish from space, is consistently greater than the surface occurrence. Stratocumulus studies have long focused on both cloud base and surface precipitation owing in part to the effects of drizzle evaporation on boundary layer

stability (e.g., Wood, 2012). It is also most natural to assess a process occurrence based on whether that process is active in the clouds at hand, and an active precipitation process in supercooled clouds will be best established from cloud base occurrence. On one hand, the difference between cloud base and surface precipitation from supercooled clouds is expected to be smaller than for ice-free stratocumulus because ice is expected to be growing during sedimentation at least immediately below liquid cloud base (in contrast to drizzle), owing to the fact that supercooled water implies a supersaturation with respect

to ice that increases with decreasing temperature. On the other hand, supercooled polar clouds can also occur at substantially higher altitudes than subtropical stratocumulus, for instance, corresponding to greater potential for sublimation before reaching the surface.

## 5 Guidance for Large-Scale Models

We suggest that these long-term ground-based statistics offer unique guidance for evaluating and improving the

representation of supercooled cloud processes in large-scale models, especially when paired with additional colocated measurements. For instance, the probability density function (PDF) of cloud base precipitation rate ($R_{\mathrm{CB}}$) from single-layer



clouds over the NSA is similar to that from all layers (Figure 3a; PDF data are provided in Table B1). Moreover, the PDF shape is largely insensitive to the cutoff altitude ($h_{min}$) up to 3 km (see Appendix B). This weak dependence of the $R_{CB}$ PDF on the cloud base height range presents a notable contrast to the strong height-dependence of satellite precipitation rate

statistics, which do not offer the context of a known liquid cloud base height (e.g., Lemonnier et al., 2020; Pettersen et al., 2018). This is likely because the underlying atmosphere's thermodynamic state has no direct influence on $R_{CB}$; in other words, below cloud base, precipitation rates are strongly influenced by the underlying supersaturation profile, as evidenced by the large differences between $R_{CB}$ and surface precipitation occurrence statistics. The PDF of $R_{CB}$ therefore offers a simple yet robust cloud process constraint, which is largely isolated from other potential thermodynamic model biases (e.g., Silber et al.,

2019b, Fig. 4). We note that the uncertainty in retrieved $R_{CB}$ values is still relatively large, but likely less than in $Z_e$-based (sixth moment) retrievals of ice crystal number concentration (a lower size distribution-moment). The general similarity of Arctic and Antarctic statistics reported here, especially at cloud base, strongly suggests that they are regionally representative at least to some degree. In other words, the basic capability of weather and climate models to reproduce a very high frequency of weak precipitation from supercooled clouds can be a useful benchmark for the performance of model physics.

Colocated measurements can furthermore serve to strengthen constraints on model processes. For instance, the ratio of a reservoir to a loss rate can be interpreted as a characteristic timescale for the loss process, and a desiccation timescale from precipitation can therefore be calculated as $\tau_{DES} = LWP/R_{CB}$ (see Bühl et al., 2016). We find that the joint histogram of $\tau_{DES}$ and cloud top temperature for single-layer NSA cases (where LWP can be reliably attributed) peaks around 6 to 9 h and –10 to -15°C (Figure 3b). Supercooled cloud occurrence is substantial in this cloud top temperature range at various levels of $Z_e$

(Figure 3c), such as both above and below –15 dBZ (a common spaceborne threshold; see Figure 1 and Figure 3c, bottom panel). $\tau_{DES}$ values shorter than the median Eulerian supercooled cloud persistence of 3 h reported over the NSA (Shupe, 2011) are more common at temperatures below -15 °C (Figure 3b, lower panel), reflecting the fact that lower $R_{CB}$ values commonly accompany lower LWP clouds (Figure 3d). Based on these statistics, we conclude that the prevalence of weak $R_{CB}$ (Figure 3a) can be important for in-cloud moisture budgets especially for low temperature and low LWP regimes that are common over

polar regions (e.g., Nomokonova et al., 2019; Shupe, 2011; Silber et al., 2018a; Zhang et al., 2010). We postulate that such fluxes are also important to below-cloud moisture budgets owing in part to the likely commonality of continued growth of ice precipitation in sub-cloud ice supersaturated conditions, which will serve to enhance moisture transport even in cases of low cloud base $Z_e$ (e.g., just above $Z_{e_{min}}$; see Appendix D).

## 6 Discussion

To our knowledge, this is the first study to report supercooled cloud base precipitation rates from an extensive sample of atmospheric profiles, including tenuous, opaque, seeded, non-seeded, single, and multi-layer clouds. Similar to Bühl et al. (2016), who studied mid-latitude geometrically thin supercooled clouds, we also evaluate the impact of cloud base ice



precipitation rates on cloud lifecycle, using ancillary measurements. We find substantially greater surface precipitation occurrence frequencies than previously reported based on lower-sensitivity spaceborne radar measurements. We posit that

such persistent ice precipitation from supercooled clouds is likely primarily supported by sustained nucleation and growth of ice crystals resulting from continuous INP activation, consistent with non-precipitating cases occurring preferentially under conditions that would generally hamper INP supply or activation rate. Persistently weak cloud base precipitation rates and precipitation loss timescales usually > 1–10 h further indicate the commonality of an INP-limited regime. Morrison et al. (2011) demonstrate that if sufficiently high ice concentrations are maintained in large-eddy simulations of a well-mixed cloud-

topped boundary layer, for instance, then surface precipitation may desiccate a low-LWP cloud layer within ~1 h. By contrast, a weakly precipitating, INP-limited regime is consistent with well-observed and widely simulated supercooled cloud case studies derived independently from several Arctic field campaigns (e.g., Fridlind & Ackerman, 2018).

The long-known commonality of ice precipitation from supercooled polar stratus and stratocumulus (e.g., Rangno & Hobbs, 2001), confirmed by these long-term measurements, suggests a role for liquid saturation as a principal gateway to polar

ice formation at temperatures between 0 and ~–38 °C (see also de Boer et al., 2011). If INP activation is the main pathway for primary ice formation, activation of immersion-mode INP is likely dominant owing to slow contact rates between droplets and interstitial aerosol particles despite cloud top phoretic enhancements (cf. Fridlind et al., 2012). Such a scenario deprecates INP activation in the deposition mode, consistent with evidence that rates are generally at least an order of magnitude weaker (e.g., Alpert et al., 2011). Supercooled stratus and stratocumulus cloud structures are generally well-reproduced by large-eddy

simulations when in-cloud ice concentrations similar to those observed are matched (e.g., Ovchinnikov et al., 2014). However, field observational constraints on both INP and ice properties have been generally insufficient to reliably predict and evaluate primary ice formation processes, and various ice multiplication processes remain highly uncertain, preventing robust conclusions from a closure approach to source attribution (Fridlind and Ackerman, 2018; Korolev et al., 2020; Lauber et al., 2018; Zipori et al., 2018).

Since temperature-dependent INP measurements over the Arctic, Antarctic, and Southern Ocean regions show large overlap with INP measurements over the NSA (Belosi et al., 2014; DeMott, 2019; Villanueva et al., 2020; Wex et al., 2019), we postulate that unremitting precipitation is likely a prevalent feature of high-latitude supercooled clouds. Precipitation loss timescales over the NSA suggest that a prevalence of weakly precipitating supercooled clouds is important for in-cloud moisture budget.

For the purposes of model evaluation, these findings underscore the importance of a "definition aware" approach (Kay et al., 2018) to enable valid comparisons between datasets obtained with differently capable instruments, or between measurements and model output while considering instrument limitations. Despite the generally high sensitivity of ground-based radar to ice precipitation, we have noted evidence that sensitivity still presents limitations to the detection of precipitation. While differing approaches to defining precipitation occurrence could have been taken in this study, we



conjecture that most would result in comparatively high occurrences relative to satellite remote-sensing capabilities, as also found for warm marine clouds (Lamer et al., 2020). Given that the great majority of clouds over both Arctic and Antarctic sites is usually precipitating, global model biases in precipitation rate could be a greater cause of error than biases in occurrence frequency (cf. Kay et al., 2018), underscoring the difference between precipitating frequently and precipitating heavily.

**7 Conclusions**

We use long-term sounding and ground-based radar measurements to characterize the properties of precipitation from supercooled clouds over North Slope of Alaska (NSA) and Antarctic (McMurdo) sites, and examine the influence of radar sensitivity on apparent precipitation occurrence. Quantitative analyses support the following conclusions:

- More than 85% (75%) of the detected supercooled cloud layers over the NSA (McMurdo) precipitate from the liquid cloud base, largely in the form of ice, and precipitation is detected close to the surface in more than 75% (50%) of supercooled cloud-containing profiles.

- Such greater prevalence of surface precipitation can be reconciled with spaceborne estimates, some of which are lesser by more than half, by considering the lower sensitivity of spaceborne radars and related precipitation detection algorithms.

- Although roughly half of the detected supercooled cloud layers are seeded by ice precipitation from aloft, precipitation occurrence is only roughly 10% lower from unseeded relative to all detected supercooled layers, indicating that supercooled clouds are commonly a source of ice in polar regions.

- Non-precipitating supercooled clouds are preferentially associated with higher temperatures, smaller LWPs, radiative shielding by overlying cloud layers, lack of in-cloud turbulence, and relatively more pristine conditions.

- An analysis of desiccation timescales based on colocated retrievals of LWP for single-layer cases over NSA indicates that the effect of persistent weak precipitation fluxes on in-cloud moisture budgets can be non-negligible owing to the commonality of low cloud LWPs.

The prevalence of precipitating polar supercooled clouds, commensurate with their frequently observed persistence, implies that large-scale models should reflect similar characteristics in order to better represent both the polar atmospheric state (e.g., phase partitioning and radiative fluxes) and cloud processes (e.g., prevalent ice nucleation and precipitation) (e.g.
Mülmenstädt et al., 2020). We suggest that supercooled cloud base precipitation rate statistics, which to our knowledge have not been a focus of model evaluation efforts to date, will be particularly valuable for evaluating and improving the representation of supercooled cloud processes in large-scale models. In contrast to evaluating precipitation rate statistics at all levels without regard for supercooled cloud boundaries, the precipitation at the cloud base level is detected in observations and evaluated based on model output before extensive growth and/or sublimation throughout the underlying atmosphere, thus

improving the robustness of the observational statistics *and* the isolation of model output from indirect biases associated with the representation of the atmospheric thermodynamic profile. Whereas current spaceborne measurements provide greater coverage, ground-based measurements can overcome some spaceborne observability limitations and provide valuable colocated observations for more detailed model process evaluation.

**Appendix A: Minimum Detectable KAZR $Z_e$**

Figure A1 shows the minimum detectable KAZR $Z_e$ over the NSA and McMurdo Station based on analysis of the full dataset discussed in Section 2. Because only the KAZR general (GE) mode properly operates below ~700 m and ~450 m above ground level (AGL) over the NSA and McMurdo, respectively, the instrument sensitivity is lower below this height. The $Z_{e_{min}}$ profiles suggest that the $Z_e$ sensitivity analysis discussed in the main text is influenced by the varying KAZR sensitivity up to ~-35 dBZ (at 4.3 km, the highest examined altitude), which implies that the actual precipitation percentage is potentially higher

for $Z_{e_{min}}$ below this value.

**Appendix B: Variability of the PDF of $R_{CB}$ Using Various Parametrizations and $h_{min}$ Values**

Figure B1 depicts the PDF of $R_{CB}$ over all sampled NSA cases using a few different $Z_e$-$R$ relationships and various values of the lowest examined (cutoff) KAZR altitude, $h_{min}$. The illustrated $Z_e$-$R$ retrievals exhibit different variance in $R_{CB}$ and show some overlap with the method used in this study, in which an $h_{min}$ value of 300 m was applied (see Table B1 for the

corresponding $R_{CB}$ PDF data). When higher $h_{min}$ values are used, the left tail of the calculated PDF narrows due to the decreasing KAZR sensitivity with increasing height (see Figure A1), but the general PDF shape including the $R_{CB}$ at the PDF mode is preserved even though the number of samples can be significantly smaller. Some narrowing of the right end of the PDF can be observed due to local differences in cloud properties at given altitudes (Figure B1). Yet, the general robustness of these PDF shapes accentuates the lack of direct dependence of $R_{CB}$ on the thermodynamic structure of the underlying

atmosphere, which could grow or sublimate the precipitating ice. Therefore, we suggest that the PDF of $R_{CB}$ may provide a simple yet robust observational constraint for testing large-scale models. We note that a separated analysis of $R_{CB}$ indicates that some seasonal changes in the PDF variance exist, but these changes are rather consistent regardless of $h_{min}$ (not shown), similar to the robustness of the annual analysis discussed above.

**Appendix C: Examples of Apparently Non-Precipitating Supercooled Cloud Layer Cases**

Figures C1, C2, and C3 provide a few examples of supercooled cloud layers detected over the NSA, in which there are some periods where the clouds apparently do not precipitate. Figure C1 portrays an hour of remote-sensing measurements from September 1, 2015, during which a non-precipitating supercooled layer is observed between 11:00 and 11:15 UTC. The



cloud top temperature is ~-5 °C during this event based on sounding measurements from the same hour. Before 11:15, there is little apparent precipitation, while in other regions and periods it appears that the KAZR GE mode is not sensitive enough to detect precipitation (note the difference between the detected hydrometeor signal above and below the dashed lines during 11:30-11:40 UTC).

Figure C2 shows a different example of a non-precipitating cloud layer observed on November 10, 2015. The topmost supercooled cloud layer (cloud top temperature of ~-20 °C) detected between 23:40-00:00 UTC appears as not precipitating because there are no detectible KAZR echoes attached to cloud base. However, precipitating ice is detected by KAZR ~150 m below cloud base. Because the relative humidity with respect to ice is above 100% between cloud base and the precipitating hydrometeors (not shown), we deduce that the cloud is actually precipitating but the backscattered KAZR moderate sensitivity (MD) mode signal is not strong enough to allow detection of these hydrometeors.

Figure C3 depicts an additional example from June 27, 2015. The multiple elevated supercooled cloud layers (~1250-2100 m; cloud top temperatures between -1 and -4 °C) are non-precipitating. The lowermost geometrically thick cloud layer (below 500 m) does not seem to precipitate as well, but this cloud layer is warm (temperatures above 0 °C), and hence, not included in our analysis.

**Appendix D: An Example of a Case with Significant Intensification of Ice-Induced $Z_e$ below Cloud Base**

Ice precipitation from supercooled clouds may occasionally produce rather low cloud base $Z_e$ (hence, largely low $R_{CB}$), but ice supersaturated conditions, aggregation, and/or riming in the underlying atmosphere can promote $Z_e$ values that are high enough to be considered as precipitation by other $Z_e$-dependent definitions such as those used in satellite retrievals. Figure D1 provides an example of such a case detected over the NSA on January 19, 2015, between 01:00 and 02:00 UTC. Lack of convergence in MWR LWP retrievals throughout this event (not shown) suggest that the actual LWP could be well below the retrieval uncertainty level. Without seeding ice directly above, it can be deduced that the origin of the ice precipitation in this case is the tenuous liquid-bearing layer detected at ~2 km, and hence, the precipitation is ultimately attributable to this layer (no liquid, no ice). The cloud base $Z_e$ is on the order of -43-(-35) dBZ, corresponding with $Z_{e_{min}}$ at 2 km over the NSA (see Figure A1). Nonetheless, continuous ice growth occurs below cloud base owing to the relatively consistent ice supersaturated conditions (RHi > 100%) indicated by the sounding profiles from 2 (4) hours prior (following) the examined period (not shown). The KAZR GE spectra profile (right panel) further supports continuous ice growth at the very least down to ~900 m, where there is an indication of vertical shear of the horizontal wind. We note that $Z_e$ in the lowest KAZR GE range gate (~170 m) ranges between -15 and +3 dBZ during the examined hour (not shown), which would have likely been considered as "certain snowfall" by satellite retrievals (if they were not affected by ground clutter at these low heights), as would most profiles up to 1,200 m (just above CPR retrievals' blind zone).



**Data Availability:**

The data used in this study, including the high spectral resolution lidar liquid cloud base height data product presented in
Appendix C are available in the ARM data archive (http://www.archive.arm.gov).

**Author Contribution:**

I.S. conceptualized the study, developed the methodology, performed the formal analysis, and prepared the manuscript. A.F. contributed to study conceptualization and manuscript preparation. J.V. contributed to study conceptualization and development of the methodology, and reviewed and edited the manuscript. A.A., G.C., and D.K. critically reviewed and edited
the manuscript.

**Competing Interests:**

The authors declare that they have no conflict of interest.

**Acknowledgments:**

We thank Tristan L'Ecuyer and Maria Cadeddu for helpful discussions. I.S. and J.V. are supported by the DOE grant DE-
SC0017981. I.S. is also supported by DOE grant DE-SC0018046. D.K. acknowledges support by the DOE grants DE-SC0020006 and DE-SC0021034 and NASA award NNX17AJ12G. A.F. and A.A. are supported by the NASA Radiation Science and Modeling, Analysis and Prediction programs. G.C. is supported by a CloudSat-CALIPSO RTOP at the NASA Goddard Institute for Space Studies.

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







**Figure 1: Precipitation occurrence over the Arctic site (left) and Antarctic site (right) as a function of radar range resolution and equivalent radar reflectivity ($Z_e$) threshold: (a,b) for all supercooled cloud layers at liquid cloud base, (c,d) for uppermost supercooled layers with no overlying hydrometeor detections, and (e,f) for surface precipitation (defined here as ~300 m above ground level) from all layer-containing columns. Symbols indicate the range resolutions and detectability thresholds for the KAZR at ~1 km above ground level, the CloudSat CPR, EarthCARE CPR, the GPM KaPR, and the CloudSat 2C-PC and 2C-SP precipitation detection algorithms (for possible liquid and ice precipitation, or certain ice precipitation).**





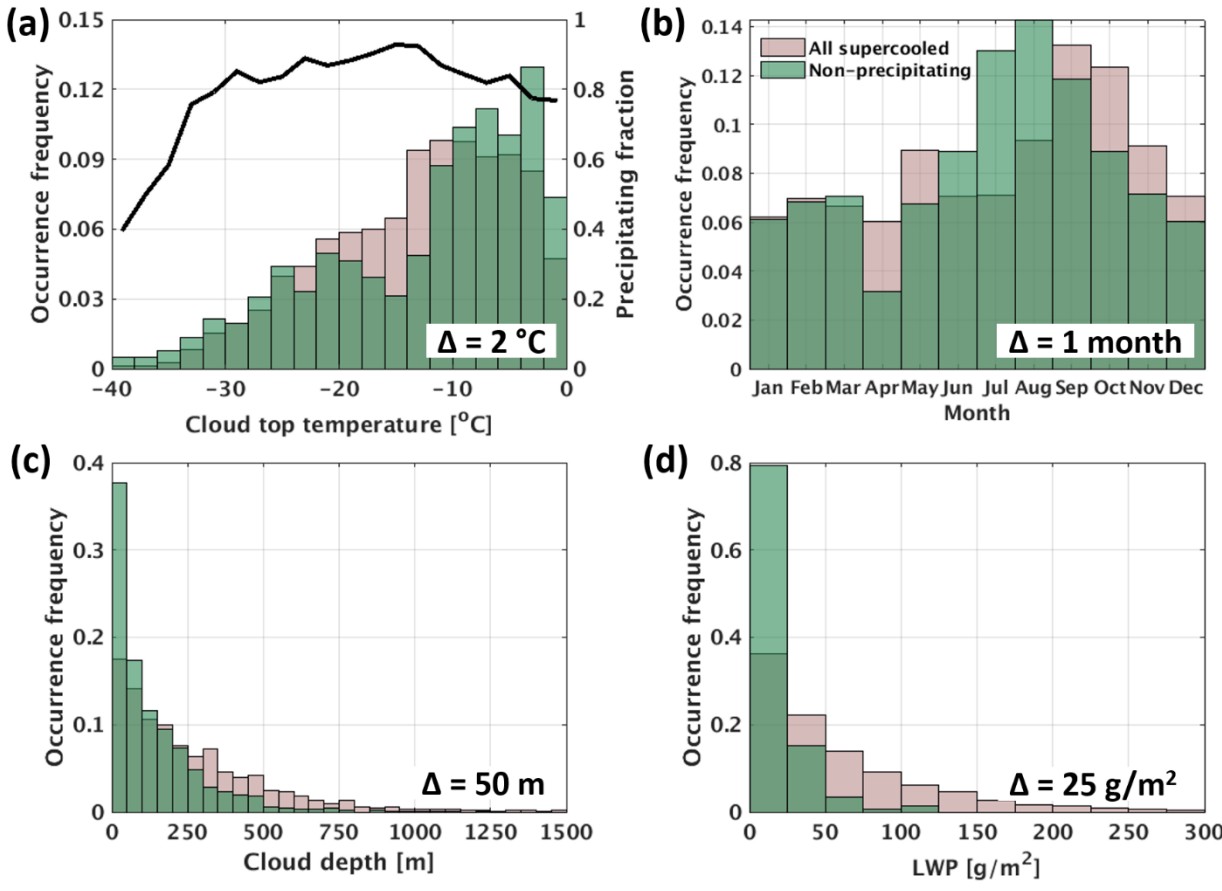

**Figure 2: Occurrence frequency histograms for supercooled cloud layers over the NSA (tan) and the non-precipitating subset (green): (a) cloud top temperature ($T_{CT}$; obtained from sounding measurements), (b) month, and (c) cloud depth for all layers, and (d) liquid water path for single-layer cases. Precipitating fractions as a function of $T_{CT}$ are shown in panel a (black line).**





**Figure 3: Precipitation characteristics over the NSA: (a) estimated cloud base precipitation rate ($R_{CB}$) probability density function (PDF; calculated over log10 of $R_{CB}$ with a logarithmic bin width of 0.5) over all sampled cases (solid + shaded) and single-layer cases (dashed). Dotted curves denote the PDF using $R_{CB}$ at its uncertainty range edges over all samples. (b,c,d) Joint histograms over single-layer cases of precipitation loss timescale $\tau_{DES}$ (see Section 5) versus $T_{CT}$, KAZR $Z_e$ immediately below supercooled cloud base versus $T_{CT}$, and LWP versus $R_{CB}$. The $\tau_{DES}$ and $R_{CB}$ histogram bins have base 10 logarithmic bin widths of 0.2 and 0.5, respectively. Integrated occurrence fractions are shown in the side panels.**





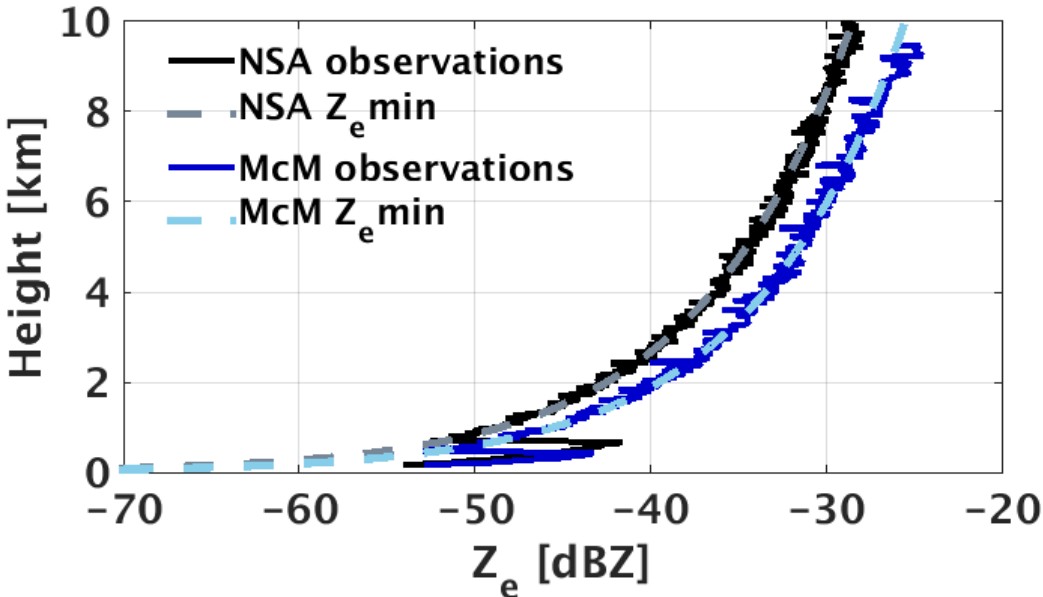

650

**Figure A1**: **Minimum detectable KAZR $Z_e$ at Utqiaġvik and McMurdo Station based on analysis of the full dataset discussed in Section 2. The smooth curves designate the theoretical minimum detectible $Z_e$ profile ( $Z_{e_{min}}$ ), using the KAZR $Z_e$ sensitivity at 1 km AGL.**



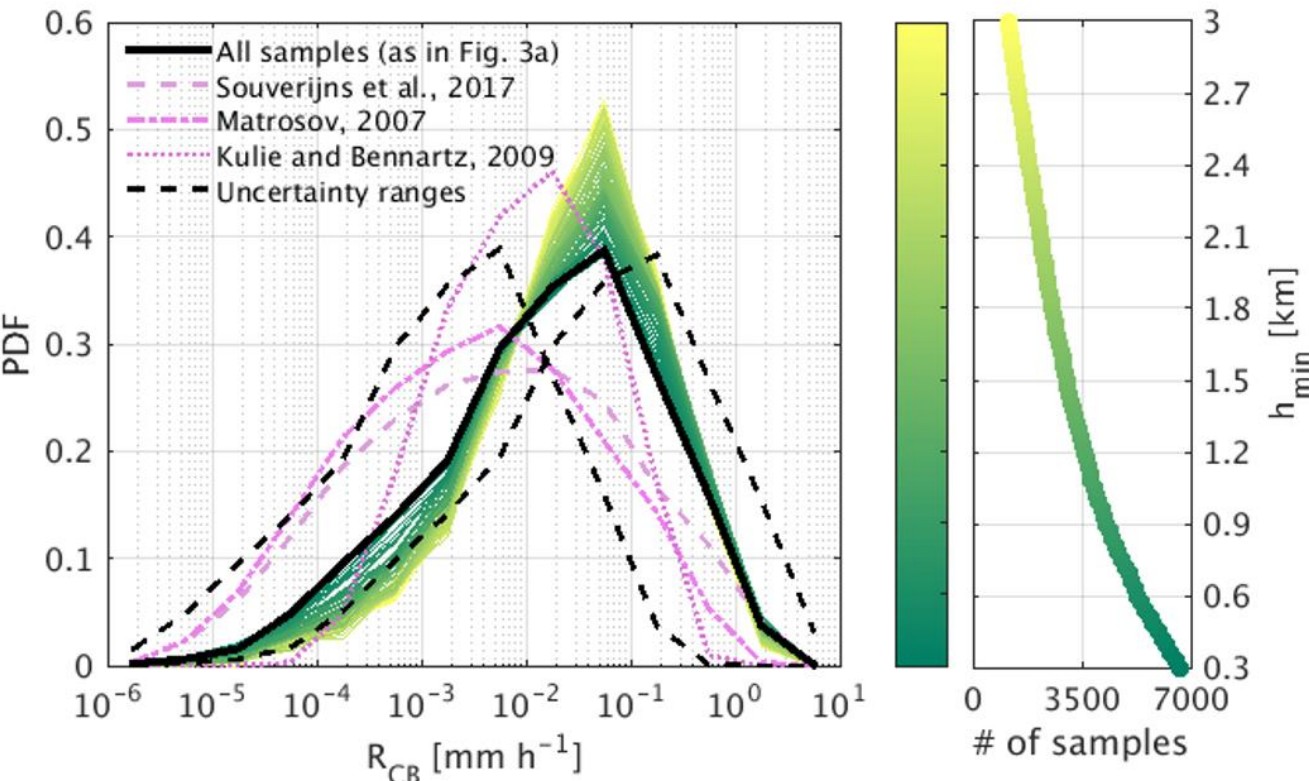

**Figure B1**: The PDF of $R_{CB}$ over all sampled NSA cases (solid black; as in Figure 3a); dashed black curves denote the PDF using $R_{CB}$ at its uncertainty range edges over all samples (as in Figure 3a). The pink curves show PDFs of $R_{CB}$ calculated using $Z_e$-$R$ relationships derived by Souverijns et al. (2017; for snowfall rate), Kulie and Bennartz (2009; for aggregates and bullet rosettes above and below -20 °C, respectively), and Matrosov (2007; for dendritic aggregates) (see legend for details). The color-scaled curves show the PDFs calculated using the same $R_{CB}$ calculation method as in the text but with various lowest examined (cutoff) KAZR altitudes ($h_{min}$); the right panel shows the number of samples for every $h_{min}$ value.



**Figure C1**: **Hour containing a non-precipitating supercooled layer between 11:00 and 11:15 UTC on September 1, 2015, over the NSA. (a) High spectral resolution lidar (HSRL; Eloranta, 2005) particulate backscatter cross-section, (b) HSRL linear depolarization ratio, (c) KAZR signal-to-noise ratio (SNR), (d) KAZR $Z_e$, and (e) KAZR cloud mask using an SNR threshold of -16 dB (see Silber et al., 2018a). Dashed horizontal white and red lines designate the height below (above) which the KAZR GE (MD) mode is used. Black or white dots designate the HSRL liquid cloud base height data product (see Silber et al., 2018c, for the algorithm description; Silber et al., 2018b, 2019c, for the data product). See Appendix C for discussion.**





**Figure C2: As in Figure C1 but for a multi-layer event on November 10, 2015. See Appendix C for discussion.**





670

**Figure C3: As in Figure C1 but for a multi-layer event on June 27, 2015. See Appendix C for discussion.**



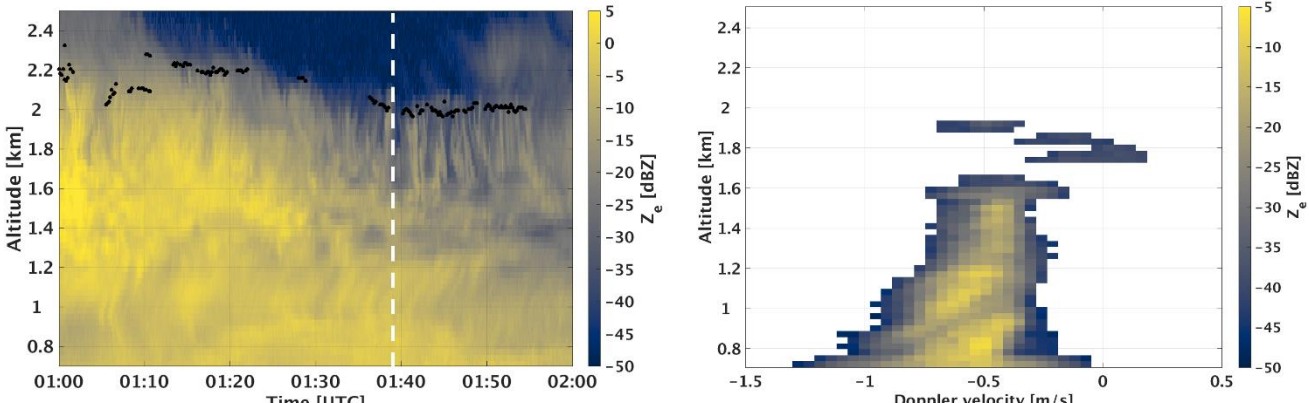

**Figure D1: (left) KAZR MD mode $Z_e$ over the NSA on January 19, 2015, between 01:00 and 02:00 UTC. Black dots designate the HSRL liquid cloud base height data product. (right) KAZR GE spectra profile at 01:39:02 UTC (designated by the dashed white line in the left panel). See Appendix D for discussion.**

675



**Table B1: PDF data illustrated in Figure 3a.**

| Bin range [$\log_{10}$(mm/h)] | | | Logarithmic bin middle $R_{CB}$ value converted to linear units [mm/h] | PDF (calculated over $\log_{10}R_{CB}$) [-log10(mm/h)] | | | |
|---|---|---|---|---|---|---|---|
| | | | | **Full dataset** | Full dataset using $R_{CB}$ at its leftmost uncertainty edge | Full dataset using $R_{CB}$ at its rightmost uncertainty edge | Single-layer subset |
| -6.00 | to | -5.50 | 0.000002 | **0.003** | 0.021 | 0.000 | 0.008 |
| -5.50 | to | -5.00 | 0.000006 | **0.008** | 0.053 | 0.003 | 0.011 |
| -5.00 | to | -4.50 | 0.000018 | **0.021** | 0.097 | 0.008 | 0.034 |
| -4.50 | to | -4.00 | 0.000056 | **0.052** | 0.142 | 0.021 | 0.072 |
| -4.00 | to | -3.50 | 0.000178 | **0.096** | 0.191 | 0.054 | 0.147 |
| -3.50 | to | -3.00 | 0.000562 | **0.141** | 0.295 | 0.100 | 0.213 |
| -3.00 | to | -2.50 | 0.001778 | **0.190** | 0.351 | 0.143 | 0.244 |
| -2.50 | to | -2.00 | 0.005623 | **0.293** | 0.386 | 0.193 | 0.358 |
| -2.00 | to | -1.50 | 0.017783 | **0.349** | 0.269 | 0.294 | 0.390 |
| -1.50 | to | -1.00 | 0.056234 | **0.384** | 0.158 | 0.355 | 0.322 |
| -1.00 | to | -0.50 | 0.177828 | **0.268** | 0.037 | 0.380 | 0.148 |
| -0.50 | to | 0.00 | 0.562341 | **0.157** | 0.001 | 0.263 | 0.040 |
| 0.00 | to | 0.50 | 1.778279 | **0.037** | 0.000 | 0.153 | 0.013 |
| 0.50 | to | 1.00 | 5.623413 | **0.001** | 0.000 | 0.032 | 0.000 |