# Peer review of "The Prevalence of Precipitation from Polar Supercooled Clouds"

_Atmospheric Chemistry and Physics, 2020_

## Referee Comment (RC1) · Anonymous Referee #1 · 15 Dec 2020

The authors analyzed multi-year data sets from NSA and McMurdo Station to investigate how often clouds are precipitating. In general, this is a very valuable and relevant contribution. The text is well written even though I think the manuscript would gain some clarity if the sentences would be less complex. The quality of the figures is very good. I enjoyed reading the paper and I have only two major comments. Given that both comments are of rather philosophical nature, I recommend the paper to be published subject to minor revisions.

Major comments:

Definition of cloud base: I'm aware that it is extremely common when dealing with mixed-phase clouds to define only the upper part as a cloud where liquid is present. Given that clouds are defined by their optical properties (the AMS glossary requires

them to be 'visible'), I'm not sure whether this is always a smart choice. Maybe, the authors are aware of that dilemma because they refer only to supercooled clouds. This makes it technically correct, but this means the authors manage to write a paper about mixed-phase clouds without using the term 'mixed-phase cloud' a single time! I would recommend to think about this choice because mixed-phase clouds is an extremely well-established term and avoiding to use it makes a paper less visible (think of people looking for relevant mixed-phase papers with Google Scholar). One way to circumvent the cloud base dilemma would be to use the term 'liquid cloud base'. Alternatively, I would recommend to explain why the term mixed-phase clouds is not used.

L128 'detectable sizes': This is actually related to the upper comment. The authors do a great job in simulating the capabilities of various space-borne sensors. But I would recommend to go one step further in the discussion: I'm missing a discussion about a meaningful threshold for precipitation. Precipitation radars (operational radars, MRRs) typically have sensitivities around 0 dBz, and this is sufficient if the goal is to measure the mass flux at the surface. A reflectivity of -50 dBz already corresponds to an almost negligible mass flux, but what if the authors had a radar with infinite sensitivity that can detect a single ice crystal falling from a supercooled cloud over the course of 15 minutes? Would the authors call this a precipitating cloud? I would expect that almost every mixed-phase cloud can generate a single precipitating crystal, i.e. is precipitating when using the definition of the authors. For the case of a radar, with infinite sensitivity, wouldn't the classification be less about distinguishing between precipitating and non-precipitating clouds and rather be about distinguishing between mixed-phase clouds (with ice formation) and purely liquid clouds? And if this is true, until what Ze value does that hold? -50 dBz? 0 dBz? Thinking about this, I have the impression the authors rather developed a classification to distinguish between mixed-phase (or ice forming) and purely liquid clouds.

Minor comments

L67: When analyzing large data sets, results can depend a lot on the choices made in

the very beginning. Why did the authors choose a 50% threshold and are the results robust to that?

L73: Averaged in linear or logarithmic space?

L90: Because the data set lengths are quite different, I would recommend to repeat the study period or use relative occurrences.

L144ff: Given the lack of INP measurements, I would recommend to trim this discussion or to include other potential mechanisms such as INP recycling.

L159: wrong figure reference

L165: The authors use 'not shown' quite often even though the discussion is interesting and would benefit from a figure.

L220f: I cannot follow the authors here, because precipitation rate is also correlated to a lower size distribution moment

L221: I would recommend to be more specific about the similarities, e.g. in L152 the differences are stressed.

L234: I'm not sure I can follow here: Is there any precipitation rate that is *not* important for the in-cloud moisture budget? In my opinion, even a precipitation rate of 0 is relevant for the budget.

Fig 1: The size of the symbols in the legend is very small

Fig 2b: I found this figure initially very confusing: First, I thought the authors show the likelihood of observing a (precipitating or non precipitating) cloud in a given month. Instead, it is how the observed clouds are distributed over the year. I would recommend to state this more clearly in the caption.

Fig. C3: It is a nice case, but I'm not sure why it is shown?

Fig. D1: The sign convention is opposite to the one reported in L83.

[Figure]

---

## Referee Comment (RC2) · Anonymous Referee #2 · 3 Jan 2021

The manuscript "The Prevalence of Precipitation from Polar Supercooled Clouds" by I. Silbert and colleagues documented an observational analysis of long-term measurements at two sites, one in Alaska and the other in Antarctica. They found that the supercooled clouds produce frequent precipitation at cloud base and most of them reach the ground. The attributed the discrepancy between their findings and the previously reported spaceborne estimates to the detection limitation of the spaceborne measurements. Finally, the authors suggest that the supercooled cloud formation is an important gateway for ice formation, and that the cloud base precipitation statistics can be used for evaluating large-scale models. Overall I find this study very interesting and the manuscript very well-prepared. The presentation is straightforward and easy to follow. The text is concise. The methods are described clearly, and the science findings

are supported by evidence and are potentially significant. Therefore, I recommend this manuscript to be published.

I only have a few comments for the authors to consider:

1. The authors did not clearly state the phase of the supercooled clouds and their precipitation. Because different mechanisms drive different clouds, it will be very helpful to explicitly discuss the phase of clouds (mixed phase, supercooled liquid, or pure ice) as well as the phase of the precipitation (liquid, ice, or both) in each analysis throughout the manuscript.

2. This study relies on $\sim$ 7 years of measurements at Alaska and $\sim$ 1 year of measurements in Antarctica. Are these results representative enough? Are there any other previous, ongoing, or future measurements that can provide further validation? The ambient environmental and meteorological conditions are not discussed in the manuscript. Do they play a role? It will be helpful if the authors provide a paragraph to discuss these aspects.

3. The sampling and the procedure for data processing are described very clearly, but it will be helpful to discuss the caveat, limitation, or consequences of the sampling or data processing procedure, especially if any of the steps (or assumptions) might affect the science conclusion. It will also be helpful for the readers to know even if the procedure will not affect any of the conclusions.

4. The discussion on the INP is a hypothesis used for explaining part of the precipitation statistics, but unfortunately the paper also indicates that there is no INL observations at the two sites and therefore the paper did not provide any evidence or analysis on INP. I am wondering if field campaign measurements can be used to bridge the gap and provide solid evidence to back your hypothesis.

5. While I understand that the authors suggested the use of precipitation statistics at cloud base for model evaluation because it is independent of atmospheric thermodynamics and can be retrieved with certain degree of confidence, it is unclear to me how it matters in climate modeling and what processes it tries to constrain. It will be much more helpful (and potentially increase the community's interest in using this new metric for evaluating models which eventually increases the scientific significance of this paper) if the authors elaborate why this metric is useful and which process(es) in models can be constrained by it.

―――――――――――――――――――――

---

## Author Comment (AC1) · 10 Feb 2021

**Author Responses**

We thank both reviewers for their valuable comments and helpful suggestions. Our responses and revisions are enumerated below.

**Reviewer #1 (Comments to Author):**

The authors analyzed multi-year data sets from NSA and McMurdo Station to investigate how often clouds are precipitating. In general, this is a very valuable and relevant contribution. The text is well written even though I think the manuscript would gain some clarity if the sentences would be less complex. The quality of the figures is very good. I enjoyed reading the paper and I have only two major comments. Given that both comments are of rather philosophical nature, I recommend the paper to be published subject to minor revisions.

Major comments:

Definition of cloud base: I'm aware that it is extremely common when dealing with mixed-phase clouds to define only the upper part as a cloud where liquid is present. Given that clouds are defined by their optical properties (the AMS glossary requires them to be 'visible'), I'm not sure whether this is always a smart choice. Maybe, the authors are aware of that dilemma because they refer only to supercooled clouds. This makes it technically correct, but this means the authors manage to write a paper about mixed-phase clouds without using the term 'mixed-phase cloud' a single time! I would recommend to think about this choice because mixed-phase clouds is an extremely well-established term and avoiding to use it makes a paper less visible (think of people looking for relevant mixed-phase papers with Google Scholar). One way to circumvent the cloud base dilemma would be to use the term 'liquid cloud base'. Alternatively, I would recommend to explain why the term mixed-phase clouds is not used.

The reviewer's comment touches on a delicate point we've been contemplating for some time now concerning how well does the definition of mixed-phase clouds stand when polar clouds (with typically low ice number concentrations and weak precipitation fluxes) are discussed. The AMS glossary's definition of a mixed-phase cloud is "*A cloud containing both water drops (supercooled at temperatures below 0°C) and ice crystals, hence a cloud with a composition between that of a water cloud and that of an ice- crystal cloud.*" In agreement with the reviewer's note, this definition raises inconsistencies concerning how do we define what should be considered as a cloud. The observational aspect of many studies such as this one introduces some additional potential inconsistencies to that definition of a mixed-phase cloud as the ground-truth about cloud-boundaries and the spatial distribution of its hydrometeors is unknown, and hence, convolved in the discretization of observed variables. We thought that writing the manuscript while only referring to the unambiguous information, that is, supercooled hydrometeor-containing air-volumes might become too confusing. Therefore, without addressing the definition of *a* cloud, in this study we only address part of this ambiguity by not using the "mixed-phase cloud" term and refer to "supercooled clouds" instead.

We added a paragraph to the Discussion Section discussing this issue (l. 280-286): "*A definitional overlap exists between precipitating supercooled clouds as defined in this study and mixed-phase clouds as defined in other studies; namely, supercooled clouds that are precipitating ice are also mixed-phase clouds. Microphysically, this overlap hinges on the rapid equilibration of supercooled cloud water with ambient vapor*

*pressure combined with the rapid growth of ice crystals at liquid saturation. However, Figure 1 shows that the diagnosed occurrence frequency of precipitating supercooled clouds, and by extension mixed-phase clouds, can depend strongly on instrument sensitivity. The probability of observing ice hydrometeors also increases with a longer duration of measurement averaging window (e.g., Figure C1). Thus, our analysis demonstrates that the observed abundance of mixed-phase clouds can vary substantially with methodology.*"

We also relate to this point of discussion in the Abstract and the Conclusions:

(l. 22-23): *"The results here also demonstrate that the observed abundance of mixed-phase clouds can vary substantially with instrument sensitivity and methodology."*

(l. 306-307): "*By extension, insofar as mixed-phase clouds are defined as supercooled clouds that are precipitating ice, the inferred abundance of mixed-phase clouds can vary substantially with instrument sensitivity and methodology.*"

Concerning the use of "cloud base", we use the term "liquid cloud base" in about 1/3 of the instances in which it is discussed. In the other instances, "liquid" is omitted from this term because in these sentences supercooled clouds are already discussed as the subject of the paragraph. Because "supercooled" refers to the liquid phase by its dictionary definition we thought that mentioning "liquid" when referring to cloud base might be redundant in these cases.

L128 'detectable sizes': This is actually related to the upper comment. The authors do a great job in simulating the capabilities of various space-borne sensors. But I would recommend to go one step further in the discussion: I'm missing a discussion about a meaningful threshold for precipitation. Precipitation radars (operational radars, MRRs) typically have sensitivities around 0 dBz, and this is sufficient if the goal is to measure the mass flux at the surface. A reflectivity of -50 dBz already corresponds to an almost negligible mass flux, but what if the authors had a radar with infinite sensitivity that can detect a single ice crystal falling from a supercooled cloud over the course of 15 minutes? Would the authors call this a precipitating cloud? I would expect that almost every mixed-phase cloud can generate a single precipitating crystal, i.e. is precipitating when using the definition of the authors. For the case of a radar, with infinite sensitivity, wouldn't the classification be less about distinguishing between precipitating and nonprecipitating clouds and rather be about distinguishing between mixed-phase clouds (with ice formation) and purely liquid clouds? And if this is true, until what Ze value does that hold? -50 dBz? 0 dBz? Thinking about this, I have the impression the authors rather developed a classification to distinguish between mixed-phase (or ice forming) and purely liquid clouds.

These are all interesting questions, which are largely addressed in our response to the previous comment and our updates to the text. The radar sensitivity issue is one of the reasons why we think that the conventional classification of polar stratiform liquid-bearing clouds into "liquid-only" and "mixed-phase" is prone to error in intercomparisons. We think that even a single precipitating ice crystal would change the definition of the associate supercooled cloud to a precipitating supercooled cloud, as we already discuss in the paragraph added in response to the previous comment.

We note that we generally do not put a lower reflectivity limit for precipitation because commonly occurring ice supersaturated conditions in the sub-cloud atmospheric profile can support the growth of initially minute ice particles (falling from cloud base) via vapor deposition to "significant" sizes and/or masses. While we agree that reflectivity

of -50 dBZ just below cloud base will be translated to $R$ values in the lower end, likely insignificant *directly* to the overlying (potentially generating) cloud moisture budget, ice particles inducing such reflectivity values could be *directly* important for the moisture budget of the underlying atmosphere and/or surface, and *indirectly*, for the moisture budget of the generating layer. Figure D2 and its associated discussion in Appendix D provide an example of such a case.

We think that the new figure (C3) added to Appendix C showing the $R_{CB}$ PDF using different $Z_{e_{min}}$ values addresses the interesting infinite radar-sensitivity question by showing that there likely is an upper limit to additional information about existing hydrometeors, which such an infinite-sensitivity radar can add.

Minor comments

L67: When analyzing large data sets, results can depend a lot on the choices made in the very beginning. Why did the authors choose a 50% threshold and are the results robust to that?

We agree that the resulting precipitation frequency might be higher due to a lower occurrence percentage threshold. Implementing the volumetric mean reflectivity thresholds for the arithmetically mean reflectivity within a temporal window mitigates some of the biases induced by binary-data averaging such as the occurrence fraction threshold (one of the two implemented conditions for precipitation detection). In the extreme case of an occurrence fraction threshold of 1, such biases do not exist, but knowing that radar data may occasionally contain artifacts that are hard to control, a value of 1 cannot be used.

We originally ran the analysis discussed in the manuscript while cycling through occurrence fraction thresholds ranging from 0.25 to 1. The light blue curve in the plot below shows the precipitation percentage per detected layer for the NSA dataset (equivalent to Figure 1a,b in the text). The behavior of the depicted curve is rather stable (quasi-linear gentle decrease), changing by merely 15% from a threshold of 0.25 to 0.95, owing in part to the stratiform nature of the clouds over both sites. Even when a threshold of 0.95 is used, a total precipitation occurrence of 76% over the NSA would not change the results and notions presented in the manuscript, apart from the actual numbers. Above 0.95, the analysis becomes less stable, and the potential artifacts receive an increasing weight over the results.

Based on this analysis, we decided to use an occurrence fraction threshold of 0.5 and mentioned in the text that (l. 71-72): *"We estimate likely biases resulting from the binary averaging (e.g., Smalley et al., 2014) on this occurrence percentage threshold to be below 10% (not shown)."*

[Figure]

L73: Averaged in linear or logarithmic space?

Added "linearly" to the text.

L90: Because the data set lengths are quite different, I would recommend to repeat the study period or use relative occurrences.

Added occurrences to that sentence (l. 92-95): *"The resulting McMurdo dataset constitutes 236 (dmin = 600 m) to 262 (dmin = 60 m) profiles with at least one supercooled layer (corresponding to a supercooled cloud occurrence frequency of 29% to 32%, respectively) whereas the NSA dataset constitutes 3,139 to 4,544 profiles (frequency of 38% to 55%, respectively), the larger range of which reflects a higher occurrence of supercooled clouds below ~1 km over the NSA (e.g., Lubin et al., 2020, Fig. 7)."*

L144ff: Given the lack of INP measurements, I would recommend to trim this discussion or to include other potential mechanisms such as INP recycling.

Added INP recycling to that discussion (l. 154-156): *"In the roughly three-quarters of cases where cloud layers are turbulent (Silber et al., 2020b), additional INP may be continuously entrained ... at cloud base via deepening of a decoupled layer (e.g., Avramov et al., 2011) or INP recycling (e.g., Solomon et al., 2015)"*

L159: wrong figure reference

Changed to "*see also Appendix C".* Thank you for pointing this error!

L165: The authors use 'not shown' quite often even though the discussion is interesting and would benefit from a figure.

We have removed three "not shown" instances where it was not essential (l. 73, l. 100, l. 392) to help reduce the impression received by the reviewer. In the case mentioned by the reviewer (radiative shielding), we originally had a panel showing the distribution of the vertical distance between overlying layers (see figure below), but eventually decided to remove this figure because we thought our point concerning radiative shielding in the scope of this paper can be conveyed using a single sentence. Some of

the other "not shown" instances originated in similar issues (e.g., the occurrence fraction threshold discussed in our response to the minor comment about l. 67).

[Figure]

L220f: I cannot follow the authors here, because precipitation rate is also correlated to a lower size distribution moment

We agree that $R_{CB}$ by its definition of being a function of IWC (third size distribution moment) is correlated with the size distribution. We argue that a higher moment order difference between the constraining and retrieved variables could imply higher uncertainties, even though the relative difference in the uncertainty depends on the PSD shape.

We toned down this sentence and added a citation to Ulbrich, 1983, which provides an interesting equivalent discussion on DSDs (l. 226-229): *"We note that because the $R_{CB}$ retrievals, described by the third moment of the size distribution (IWC) weighted by $V_D$, are based on radar reflectivity measurements (sixth moment), the uncertainties associated with the resultant $R_{CB}$ values are likely to be smaller than in equivalent $Z_e$-based retrievals of ice crystal number concentration (zeroth moment) (see Ulbrich, 1983)."*

L221: I would recommend to be more specific about the similarities, e.g. in L152 the differences are stressed.

Change this sentence to (l. 231-232): *"... the general similarity of Arctic and Antarctic precipitation occurrence reported here at cloud base strongly suggests that ..."*

L234: I'm not sure I can follow here: Is there any precipitation rate that is *not* important for the in-cloud moisture budget? In my opinion, even a precipitation rate of 0 is relevant for the budget.

We mainly refer to cloud sinks here compared to typical polar cloud lifecycles. We updated that sentence accordingly (l. 243-244): *"Based on these statistics, we conclude that the prevalent weak $R_{CB}$ (Figure 3a) can be important cloud moisture sinks …"*

Fig 1: The size of the symbols in the legend is very small

Fixed.

Fig 2b: I found this figure initially very confusing: First, I thought the authors show the likelihood of observing a (precipitating or non precipitating) cloud in a given month. Instead, it is how the observed clouds are distributed over the year. I would recommend to state this more clearly in the caption.

We understand the confusion and modified the figure caption to address this issue: *"(a) Cloud top temperature ($T_{CT}$; obtained from sounding measurements), (b) month, (c) cloud depth, and (d) liquid water path occurrence frequency histograms for supercooled cloud layers over the NSA (tan) and the non-precipitating subset (green) …"*

Fig. C3: It is a nice case, but I'm not sure why it is shown?

We thought that this case would be a nice example of the radiative shielding effect as we mention in the text. However, we agree that it is similar in essence to Figure C1 (non-precipitating cloud temperature closer to 0 °C), and hence, we replaced it with the updated Figure C3 discussed in our response to the second major comment.

Fig. D1: The sign convention is opposite to the one reported in L83.

The sign convention in l. 83 (now l. 86) was wrong. We change it from "positive" to "negative" and also updated l. 104-105 to remain consistent: *"To estimate precipitation rate (R) immediately below cloud base … we simply multiply $V_D$ magnitudes (when $V_D$ is pointing towards the surface) by retrieved ice water content (IWC) following Hogan et al. (2006)."*

**Reviewer #2 (Comments to Author):**

The manuscript "The Prevalence of Precipitation from Polar Supercooled Clouds" by I. Silbert and colleagues documented an observational analysis of long-term measurements at two sites, one in Alaska and the other in Antarctica. They found that the supercooled clouds produce frequent precipitation at cloud base and most of them reach the ground. The attributed the discrepancy between their findings and the previously reported spaceborne estimates to the detection limitation of the spaceborne measurements. Finally, the authors suggest that the supercooled cloud formation is an important gateway for ice formation, and that the cloud base precipitation statistics can be used for evaluating large-scale models. Overall I find this study very interesting and the manuscript very well-prepared. The presentation is straightforward and easy to follow. The text is concise. The methods are described clearly, and the science findings are supported by evidence and are potentially significant. Therefore, I recommend this manuscript to be published.

I only have a few comments for the authors to consider:

1.  The authors did not clearly state the phase of the supercooled clouds and their precipitation. Because different mechanisms drive different clouds, it will be very helpful to explicitly discuss the phase of clouds (mixed phase, supercooled liquid, or pure ice) as well as the phase of the precipitation (liquid, ice, or both) in each analysis throughout the manuscript.

The clouds we discuss in the text all contain liquid droplets following the dictionary definition of supercooling. To address this comment and another comment received from Reviewer #1, we added some text to the Discussion Section mentioning the overlap between commonly-termed mixed-phase clouds and the precipitating supercooled clouds discussed in this study (l. 280-286): "*A definitional overlap exists between precipitating supercooled clouds as defined in this study and mixed-phase clouds as defined in other studies; namely, supercooled clouds that are precipitating ice are also mixed-phase clouds. Microphysically, this overlap hinges on the rapid equilibration of supercooled cloud water with ambient vapor pressure combined with the rapid growth of ice crystals at liquid saturation. However, Figure 1 shows that the diagnosed occurrence frequency of precipitating supercooled clouds, and by extension mixed-phase clouds, can depend strongly on instrument sensitivity. The probability of observing ice hydrometeors also increases with a longer duration of measurement averaging window (e.g., Figure C1). Thus, our analysis demonstrates that the observed abundance of mixed-phase clouds can vary substantially with methodology.*"

We also relate to this point of discussion in the Abstract and the Conclusions:

(l. 22-23): *"The results here also demonstrate that the observed abundance of mixed-phase clouds can vary substantially with instrument sensitivity and methodology."*

(l. 306-307): "*By extension, insofar as mixed-phase clouds are defined as supercooled clouds that are precipitating ice, the inferred abundance of mixed-phase clouds can vary substantially with instrument sensitivity and methodology.*"

Our focus in this study is mostly on the precipitating ice hydrometeors because when present, they typically dominate the radar signal over potential liquid drops (e.g., Silber et al., 2019, their fig. 3). As we note in the manuscript (l. 144-147): *"The precipitation detected with KAZR may be liquid or ice phase. However, since in these datasets $Z_e$ usually increases from cloud base to some distance below (see Appendix D), indicating*

*continued ice growth during sedimentation rather than drizzle or rain evaporation, we infer that ice is the dominant precipitation form"*

To further address this comment, we added a figure to Appendix D showing the fraction of cases with increasing $Z_e$ below cloud base. We think that while unmasking potential supercooled drizzle signals from these and other datasets is a highly valuable goal, we expect it to be a challenging project for various reasons, and hence, think it is beyond the scope of this study.

2. This study relies on ~ 7 years of measurements at Alaska and ~ 1 year of measurements in Antarctica. Are these results representative enough? Are there any other previous, ongoing, or future measurements that can provide further validation? The ambient environmental and meteorological conditions are not discussed in the manuscript. Do they play a role? It will be helpful if the authors provide a paragraph to discuss these aspects.

The two examined sites are representative of the essential physics that are characteristic in polar regions (low aerosol particle concentrations, prevalent stratiform mixed-phase cloud, longwave radiation dominance, etc.). The analysis is anchored on one of the highest quality ground-based polar data streams that we have in the community (at the NSA Arctic site), which enables the calculation of robust statistics; the shorter data set over Antarctica paints a remarkably similar picture. Thus, even though these are only two sites, they provide a richer perspective on precipitation and driving processes in supercooled clouds.

Both examined sites represent relatively contrasting polar conditions (drier, colder, and more pristine atmosphere over the Antarctic site), yet the results based on these datasets are qualitatively similar. Other equivalent Arctic measurements currently take place in Ny-Ålesund, Svalbard, but based on the contrast between the Arctic and Antarctic sites, which are starker than the differences between the NSA and Ny-Ålesund (e.g., Shupe et al., 2011), this dataset would likely produce equivalent statistics.

We agree that the role of regional and meteorological conditions on our results was only mentioned with regards to the INP sources. Therefore, we elaborated on the regional cloud temperature differences as well (l. 154-158): *"The overall differences in detected Arctic versus Antarctic precipitation frequency (Figure 1) are likely influenced by geographical INP variability associated with both long-range transport and local source regions (e.g., Vergara-Temprado et al., 2018), as well as by the differing cloud temperatures (e.g., Lubin et al., 2020; Scott and Lubin, 2016), which impact INP activation (e.g., Kanji et al., 2017; Knopf et al., 2018)."*

We also mention now the fact that the results are qualitatively similar in spite of the different climatology between the two sites (l. 229-232): *"The Arctic and Antarctic sites represent relatively contrasting polar conditions, with a drier, colder, and more pristine atmosphere over McMurdo (e.g., Bromwich et al., 2012; Lubin et al., 2020; Shupe et al., 2011; Silber et al., 2018a). Despite these atmospheric state differences between the two sites, Tthe general similarity of Arctic and Antarctic precipitation occurrence statistics reported here, especially at cloud base, strongly suggests that they are regionally representative at least to some degree."*

3. The sampling and the procedure for data processing are described very clearly, but it will be helpful to discuss the caveat, limitation, or consequences of the sampling or data processing procedure, especially if any of the steps (or assumptions) might

affect the science conclusion. It will also be helpful for the readers to know even if the procedure will not affect any of the conclusions.

We discuss the limitations and caveats in our methodology throughout the text. For example, with regards to using the mean Doppler velocity in the precipitation rate calculations because we lack mass-weighted velocity calculations (l. 104-107): *"To estimate precipitation rate (R) ... because a reliable retrieval of mass-weighted fall speed is not available, we ... Using this method, we do not apply any ice habit property constraints on the observations, which span the full heterogeneous freezing temperature range..."*

In addition, we mention various uncertainty estimates, each of which is based on a separate analysis we performed and left out of the text, and hence, to keep the storyline consistent we stated "not shown" at various locations in the text (e.g., when discussing potential biases associated with hydrometeor occurrence fraction threshold – l. 72, the errors in the $Z_e$ averaging – l. 83, the robustness of $R_{CB}$ in different seasons – l. 346).

To further address this comment, we added some additional text discussing the limitations of the sounding RH threshold method for liquid-bearing air-volume detection (l. 63-67): *"This method shows good agreement (in more than 90% of cases) with independent retrievals from lidar measurements (see Silber et al., 2020b, Fig. S1) ... . While some liquid-bearing air volumes may be missed where the reported in-cloud RH is below the 95% threshold, these uncommon occurrences have little influence on the results statistically."*

We also note in the caption of Figure 2 the limited number of samples below a temperature of -34 °C (influencing the precipitation occurrence statistics) and added the number of samples to the caption of Figure 3a ($R_{CB}$ histogram) to emphasize the high number of samples relative to the number of histogram bins. While working on these updates, we were able to locate and fix an inaccurate number of samples previously given in Figure B1.

4. The discussion on the INP is a hypothesis used for explaining part of the precipitation statistics, but unfortunately the paper also indicates that there is no INL observations at the two sites and therefore the paper did not provide any evidence or analysis on INP. I am wondering if field campaign measurements can be used to bridge the gap and provide solid evidence to back your hypothesis.

We think that long-term in-situ field campaigns might be able to bring us a few steps closer to direct observational validation of our observationally-based conclusions supported by the literature. For example, long-term tethered balloon deployments could potentially provide numerous cloud and aerosol (INP) profiles (e.g., Creamean et al., 2018). We think that these balloons are the best candidates for such an effort, largely because they are generally not limited by flight hours (as in aircraft field campaigns).

We added a citation to Creamean et al. (2018) and Creamean et al. (2021) to the text (l. 161): *"Profiles of INP or aerosol properties (e.g., Creamean et al., 2018, 2021) are unfortunately not retrievable from the available McMurdo and NSA measurements, but ..."*

5. While I understand that the authors suggested the use of precipitation statistics at cloud base for model evaluation because it is independent of atmospheric thermodynamics and can be retrieved with certain degree of confidence, it is unclear to me how it matters in climate modeling and what processes it tries to constrain. It will be much more helpful (and potentially increase the community's interest in

using this new metric for evaluating models which eventually increases the scientific significance of this paper) if the authors elaborate why this metric is useful and which process(es) in models can be constrained by it.

As we suggest in the text, we think that this metric is useful because it is independent of the underlying atmospheric state, thereby allowing us to break the evaluation of ice precipitation process representation in large-scale models into two sub-categories: first, the cloud base precipitation rate, the outcome of ice nucleation, growth, and precipitation fall speed within the cloud, and second, additional growth or sublimation below the cloud, which strongly depends on how well a model represents the underlying atmospheric state. Our suggested metric addresses the first sub-category.

We reworded the conclusion section sentences where we list the relevant cloud processes to make the text clearer (l. 322-327): *"The prevalence of precipitating polar supercooled clouds, commensurate with their frequently observed persistence, implies that large-scale models should reflect similar characteristics in order to better represent both the polar atmospheric state (e.g., phase partitioning and radiative fluxes) and cloud processes (e.g., prevalent ice nucleation, growth, and precipitation) (e.g. Mülmenstädt et al., 2020). We suggest that supercooled cloud base precipitation rate statistics, which to our knowledge have not been a focus of model evaluation efforts to date, will be particularly valuable for evaluating and improving the representation of these supercooled cloud processes in large-scale models."*